# The Immunosuppressive Receptor CD32b Regulation of Macrophage Polarization and Its Implications in Tumor Progression

**DOI:** 10.3390/ijms25179737

**Published:** 2024-09-09

**Authors:** Hong-Jing Chuang, Ying-Yin Chen, Yi-Da Chung, Evelyn Huang, Cadence Yoshang Huang, Jrhau Lung, Chung-Yu Chen, Hui-Fen Liao

**Affiliations:** 1Department of Pathology, National Taiwan University Hospital Yunlin Branch, Yunlin 640, Taiwan; y05092@ms1.ylh.gov.tw; 2Department of Biochemical Science and Technology, National Chiayi University, Chiayi 600, Taiwan; s1103354@mail.ncyu.edu.tw; 3Department of Internal Medicine, National Taiwan University Hospital Yunlin Branch, Yunlin 640, Taiwan; sally771012@gmail.com; 4Cell and Molecular Biology, Northeastern University, Boston, MA 02115, USA; eve031017@gmail.com; 5Taipei American School, Taipei 111, Taiwan; kk326h@gmail.com; 6Department of Medical Research and Development, Chang Gung Memorial Hospital, Chiayi Branch, Chiayi 613, Taiwan

**Keywords:** FC gamma receptor CD32b, macrophage polarization, classically activated macrophage (M1), alternatively activated macrophage (M2), tumor stage

## Abstract

Macrophages, pivotal components of the immune system, orchestrate host defense mechanisms in humans and mammals. Their polarization into classically activated macrophages (CAMs or M1) and alternatively activated macrophages (AAMs or M2) dictates distinct functional roles in immunity and tissue homeostasis. While the negative regulatory role of CD32b within the FC gamma receptor (FCγR) family is recognized across various immune cell types, its influence on macrophage polarization remains elusive. This study aimed to elucidate the regulatory role of CD32b in macrophage polarization and discern the differential expression markers between the M1 and M2 phenotypes following CD32b siRNA transfection. The results revealed a decrease in the CD32b levels in lipopolysaccharide (LPS)-treated M1 and an increase in interleukin-4 (IL-4)-treated M2 macrophages, as observed in macrophage Raw264.7 cells. Furthermore, CD32b siRNA transfection significantly downregulated the M2 markers (IL-10, VEGF, Arg-1, and STAT6), while upregulating the M1 markers (IL-6, NF-κB, NOS2, and STAT1) in the Raw264.7 cells. Similar findings were recapitulated in macrophage-rich adherent cells isolated from mouse spleens. Additionally, the cytopathological analysis of pleural effusions and ascitic fluids from patients with cancer revealed a positive correlation between advanced tumor stages, metastasis, and elevated CD32b levels. In conclusion, this study highlights the regulatory influence of CD32b in suppressing M1 expression and promoting M2 polarization. Moreover, heightened M2 activation and CD32b levels appear to correlate with tumor progression. A targeted CD32b blockade may serve as a novel therapeutic strategy to inhibit M2 macrophage polarization and is promising for anti-tumor intervention.

## 1. Introduction

Macrophages, pivotal regulators within the immune system, exhibit diverse polarization states characterized by distinct gene and protein expression profiles [1]. The two primary macrophage phenotypes are classically activated macrophages (CAMs or M1) and alternatively activated macrophages (AAMs or M2), each contributing differently to inflammatory responses and tissue homeostasis [2,3]. Previous research has unveiled the significance of macrophage polarization in diseases such as atherosclerosis, where modulating the macrophage phenotypes holds therapeutic promise for improving plaque stability [2,4]. Additionally, in tumor biology, the dichotomy between M1 anti-tumor macrophages and M2 tumor-associated macrophages (TAMs) underscores the pivotal role of macrophage polarization in tumor progression [5].

Fcγ receptors (FcγRs) bind to the Fc region of immunoglobulin G (IgG) molecules and play a key role in immune responses. The FcγR family includes FcγRI (CD64), FcγRII (CD32), and FcγRIII (CD16), which regulate the immune functions in various cells, like macrophages, monocytes, neutrophils, B cells, and natural killer cells. Most FcγRs activate immune responses, but FcγRIIb (CD32b) is the only receptor in the family that inhibits them [6]. Notably, other studies have observed significantly reduced CD32b expression on the macrophages in patients with autoimmune diseases, particularly among Asian populations, with defects in CD32b associated with systemic lupus erythematosus (SLE) susceptibility [7,8,9]. Moreover, immune-suppressive agents like dexamethasone have been shown to upregulate the CD32b mRNA and protein levels, highlighting their potential as a therapeutic target [10]. In mouse models, CD32b-targeting antibodies have demonstrated efficacy in inhibiting tumor growth, underscoring the role of CD32b in cancer immunotherapy [11,12]. Notably, high CD32b expression levels in certain patients with non-Hodgkin’s lymphoma have been associated with resistance to rituximab treatment, suggesting a potential mechanism for immune evasion in cancer [13].

Despite its established role as a negative regulator of macrophage activation [14], the precise influence of CD32b on macrophage polarization and its implications in tumor progression remain elusive. Therefore, this study aims to elucidate the regulatory role of CD32b in macrophage polarization and discern differential expression markers between the M1 and M2 phenotypes through CD32b siRNA transfection. The initial investigations focus on assessing CD32b expression in macrophage Raw264.7 cells treated with lipopolysaccharide (LPS) for M1 polarization and interleukin-4 (IL-4) for M2 polarization. Subsequently, the markers for M1 and M2 expression in both the macrophage cell line and primary macrophages from mice following CD32b siRNA treatment are analyzed. Finally, the correlations between the CD32b levels in patients with cancer, the tumor stage, and metastasis are evaluated. The findings from this study may elucidate the role of CD32b in macrophage polarization and offer insights into its potential significance in the management of patients with cancer.

## 2. Results

### 2.1. CD32b Expression in Macrophages M1 and M2

Based on previously published research [15], the macrophage Raw264.7 cells were treated with PBS (M0), LPS (M1), and IL-4 (M2) to induce the polarization of macrophages, as shown in Figure 1. The treated cells were assayed for CD32b gene expression by qPCR, showing that the relative expressions of CD32b were decreased and increased in the M1 and M2 cells, respectively (Figure 1A). The results demonstrated that CD32b was downregulated in M1 and upregulated in the M2 cells. Figure 1B also demonstrates that nitric oxide (NO) production was increased and decreased in the M1 and M2 cells, respectively, confirming that the induction of macrophage polarization occurred. Figure 1C shows the fluorescent staining of the labeled cell surface antigen CD32b. The results demonstrate that the CD32b protein levels on the cell surface were decreased in M1 and increased in the M2 cells, respectively. It is suggested that CD32b may be involved in reducing M1 expression and regulating macrophage polarization toward M2.

### 2.2. CD32b Expression in Macrophages Transfected with CD32b siRNA

Figure 1 has proven that CD32b regulates the macrophages to move towards M2 polarization. Therefore, siRNA technology was further used to inhibit CD32b and further confirm the role of CD32b in macrophage polarization. As shown in Figure 2, CD32b siRNA was transfected to the macrophage Raw264.7 cell line and primary macrophages from the mouse spleens. Using RT-qPCR to detect the relative expression of CD32b, the Raw264.7 cells transfected with CD32b siRNA demonstrated the significantly increases expression of CD32b mRNA (Figure 2A). The assay of the primary macrophages from the mouse spleens also demonstrated the similar results (Figure 2B). The results in Figure 2 show that CD32b expression was downregulated by CD32b siRNA transfection in both the macrophage cell lines and the primary culture of macrophages.

### 2.3. M1 Marker Expression in Macrophage Raw264.7 Cells and Primary Macrophages from the Mouse Spleens Transfected with CD32b siRNA

Activated M1 macrophages can express markers, such as IL-6, NF-κB, NOS2, and STAT1 [15]. To further determine the regulatory effect of CD32b siRNA transfection on macrophage M1 markers, an RT-qPCR assay was performed, as shown in Figure 3, and the primers are listed in Table 1. In Figure 3A, the expression levels of IL-6, NF-κB, NOS2, and STAT1 were increased in the CD32b siRNA-transfected Raw264.7 cells. Similar to Figure 3A, the relative expression of M1 markers was also increased in the CD32b siRNA-transfected primary macrophages (Figure 3B). These results demonstrate that the M1 markers IL-6, NF-κB, NOS2, and STAT1 were upregulated by the cells with CD32b siRNA transfection.

### 2.4. M2 Marker Expression in Macrophage Raw264.7 Cells and Primary Macrophages from the Mouse Spleens Transfected with CD32b siRNA 

Activated M2 macrophages can express markers, such as IL-10, VEGF, Arg-1, and STAT6 [15]. The RT-qPCR assay was performed, as shown in Figure 4, and the primers are listed in Table 1. As shown in Figure 4A, the expression levels of IL-10, VEGF, Arg-1, and STAT6 were decreased in the CD32b siRNA-transfected macrophage Raw264.7 cells. Similar to Figure 4A, the number of M2 markers was also decreased in the CD32b siRNA-transfected primary macrophages (Figure 4B). These results demonstrate that the M2 markers IL-10, VEGF, Arg-1, and STAT6 were downregulated by the macrophages transfected with CD32b siRNA.

### 2.5. CD32b Expression in Human Subjects

The present results (Figure 1, Figure 2, Figure 3 and Figure 4) suggest that CD32b may regulate macrophage polarization toward M2. The previous studies have shown that M2 may be a tumor-associated macrophage (TAM) and promote tumor growth [3,5]. Therefore, this study further assayed the CD32b expression in human subjects. Table 2 lists the collected information of 32 patients with cancer who developed pleural effusion and ascitic fluid, including the tumor stage T0–T1 (*n* = 6) and T2–T4 (*n* = 26), as well as non-metastasis (*n* = 6) and distant metastasis (*n* = 26). In Figure 5, the cytopathological sections of the patient’s pleural effusion and ascitic fluid were collected and assayed by IHC staining. Immune cells such as monocytes/macrophages, lymphocytes, and polymorphonuclear cells (PMN) were observed in these pathological sections, and the macrophages (CD68) and B cells (CD19) were the main cells expressing CD32b among them. CD32b is an important inhibitory receptor on both B cells and macrophages, which suppresses immune function. As shown in Figure 5A, the ratio of CD32b/(CD19+CD68) between tumor stages T0–T1 (*n* = 6) and T2–T4 (*n* = 26) showed that the more advanced tumor stages T2–T4 expressed more CD32b than that in stages T0–T1 (*p* < 0.05). In Figure 5B, the patients with cancer with distant metastasis have higher CD32b levels than those in the non-metastasis group, but there was no significant difference. Additionally, there were no statistical differences in age and gender. In comparing Figure 5C and Figure 5D, the IHC staining of the cytopathological sections from the people with stages T0–T1 (six patients) and T2–T4 (eight selected patients), respectively, shows that the CD32b levels were much higher in the later tumor stages. The expression levels of CD19 (B cells) and CD68 (macrophages) were higher in the T2–T4 stage tumors than in the T0–T1 stage ones, indicating that the number of immune cells increased in the later stages of the tumor. However, the high expression level of CD32b at the T2–T4 stage (Figure 5D) indicates that immune cells initiate the CD32b-mediated inhibitory signaling pathway, which may cause anti-tumor immunity to be blocked.

## 3. Discussion

CD32b, a member of the FCγR family, serves as a glycoprotein receptor binding the Fc region of IgG antibodies. Among mammals, five low-affinity FcγRs exist, comprising one inhibitory receptor, CD32b (FcγRIIb), and four stimulatory receptors, CD32a (FcγRIIa), CD32c (FcγRIIc), CD16a (FcγRIIIa), and CD16b (FcγRIIIb) [16,17,18]. Research has demonstrated the expression of various low-affinity receptors, including CD16a, CD32a, CD32b, and CD32c in human natural killer (NK) cells, along with CD16 and CD32b in mouse NK cells [19,20,21]. Notably, in HER2-targeted breast cancer therapy, enhancing NK cell-mediated antibody-dependent cellular cytotoxicity (ADCC) via activating FcγRIIIa (CD16a) and macrophage-mediated antibody-dependent cellular phagocytosis (ADCP) through activating FcγRIIa (CD32a) and FcγRIIIa, alongside diminishing binding to inhibitory CD32b (FcγRIIb), has demonstrated enhanced cancer cell eradication [19,20]. The genetic deficiency of CD32b has been shown to enhance active FcγR activity, promoting monoclonal antibody (mAb) immunotherapy efficacy [21]. Additionally, receptors such as TIM-3, PD-1, CD32b, and CD200R function to inhibit macrophage activation, contributing to immune tolerance [14,22].

In our study, we induced macrophage polarization in Raw264.7 cells using PBS, LPS for M1 polarization, and IL-4 for M2 polarization, respectively. The results indicated the downregulation of CD32b levels in M1 and upregulation in the M2 cells, as confirmed by both a RT-qPCR assay and cell surface antigen fluorescence staining (Figure 1). Additionally, M1 and M2 polarization was validated by increased and decreased NO production in the M1 and M2 cells, respectively.

The inhibitory mechanism of CD32b involves the recognition of the Fc portion of IgG, leading to the recruitment of phosphatases such as SHP or SHIP into the intracellular ITIM domains of the receptor, subsequently blocking macrophage phagocytosis [14]. Our study further demonstrated the downregulation of CD32b expression following CD32b siRNA transfection in both the Raw264.7 cells and the primary macrophages from the mouse spleen cultures (Figure 2). Moreover, LPS was used to induce the proinflammatory cytokines IL-6 and TNF-α, as well as stimulate the expression of the signaling molecule NF-κB, NO synthase NOS2, and the transcription factor STAT1 in M1 macrophages [23]. However, IL-4 induced STAT6-mediated M2 polarization, and this upregulated molecules such as anti-inflammatory cytokine IL-10, angiogenesis factor VEGF, arginase Arg-1, and transcription factor STAT6 [24]. A similar model to the findings from a previous study on the natural product *Mahonia oiwakensis* and its active compounds, berberine and palmatine, modulated macrophages by suppressing M1-mediated inflammation and enhancing M2-mediated VEGF secretion and STAT6 expression [15]. Our results showed the downregulation of M2 markers (IL-10, VEGF, Arg-1, and STAT6) and the upregulation of M1 markers (IL-6, NF-κB, NOS2, and STAT1) following the CD32b siRNA transfection of both the macrophage cell line and the primary macrophages (Figure 3 and Figure 4). Additionally, tumor-associated macrophages (TAMs) predominantly play an M2-like tumor-promoting role and regulate various malignant effects, such as angiogenesis, immune suppression, and tumor metastasis [25,26]. Our results suggest that CD32b in M2 macrophage polarization may have potential as an indicator of immunosuppression in cancer and as an immunotherapeutic target for CD32b inhibition.

In B cells, CD32b exerts inhibitory signals to maintain immune homeostasis and prevent B cell activation against self-antigens. Consequently, stimulating CD32b activation emerges as a novel strategy for autoimmune and inflammatory disease treatment [27]. Although few studies have explored the relationship between CD32b and cancer, the mounting evidence suggests its involvement in tumor immune evasion. In a mouse model, CD32b contributed to a suppressive tumor microenvironment [28,29]. Furthermore, previous research has highlighted the clinical significance of CD32b in promoting M2 activation, with M2-mediated TAM activation implicated in tumor growth promotion through various mechanisms, such as angiogenic regulation, immune suppression, and metastasis facilitation [3,5]. In our study, the analysis of cytopathological sections from patients with cancer revealed higher CD32b expression levels in the advanced tumor stages (T2–T4, *n* = 26) compared to those of the earlier stages (T0–T1, *n* = 6), suggesting a potential role for CD32b in tumor progression (Figure 5, Table 2). However, no significant difference in the CD32b levels was observed between the patients with cancer with and without distant metastasis. A previous study revealed tumor and TAM interaction in the metastatic microenvironment mediated by tumor-derived exosomes that affect colorectal cancer liver metastasis [30]. Hou et al. (2021) reported that the higher-level expression of regulatory T cell (Tregs)-derived soluble fibrinogen-like protein 2 (sFGL2) and its receptor CD32b may induce macrophages toward the pro-repair markers CD163 and CD206 via the SHP2-ERK1/2-STAT3 signaling pathway, which is involved in the progression of endometriosis [31]. Modak et al. (2022) showed that alternatively activated CD206^+^ M2a macrophages efficiently cross-present soluble tumor-associated antigens (TAAs), which is a leading tumor antigen-directed cytotoxic CD8^+^ T cell response and type I IFN signaling mechanism, a key aspect of anti-tumor immunity [32]. Therefore, it is important to reconfirm the molecular mechanism of the CD32b signaling pathway in macrophages and other immune cells and the clinical immunohistochemical expression of CD32b in solid tumors of different organs in future studies to further analyze the correlation with prognostic parameters and metastasis and to use the chronic inflammation of the organ as the control.

## 4. Materials and Methods

### 4.1. Cells and Animals

Murine macrophage RAW264.7 cells were obtained from American Type Culture Collection (ATCC, Manassas, VA, USA) and maintained in Dulbecco’s Modified Eagle’s medium (DMEM; Gibco, Grand Island, NY, USA) supplemented with 10% fetal bovine serum (FBS), 2 mM L-glutamine, 1.5 g/L sodium bicarbonate, 0.1 mM nonessential amino acids, and 1.0 mM sodium pyruvate. For collecting normal macrophages from primary culture, male BALB/c mice (6–8 weeks old) were purchased from the National Laboratory Animal Center (Taipei, Taiwan). The animal use protocol was reviewed and approved by the institute of animal care and use committee (IACUC) of National Chiayi University, Taiwan (Approval no. 109011), and all the mice were bred under specific pathogen-free conditions. All the animal experiments were carried out in accordance with the guidelines in the NIH Guide for the Care and Use of Laboratory Animals (DHHS publication No. NIH 85-23, revised 1996). The mice were sacrificed, and the spleen organs were removed. RPMI 1640 medium 10 mL was added to the spleen organs, homogenized the tissue into single cells, and dispersed in a suspension by using a 1 mm metal sieve. The isolated spleen cells (5 × 10^5^ cells/mL) were cultured at 37 °C in RPMI 1640 medium containing 10% FBS for 2 h, and the adherent cells were a macrophage-rich fraction. A rubber scraper was used to collect the cells and suspend them in the culture medium. The method to induce macrophage polarization was performed following established protocols outlined in the previously published studies [15]. Briefly, Raw264.7 cells and primary macrophages isolated from the mouse spleen cells were subjected to treatment with phosphate-buffered saline (PBS, M0), lipopolysaccharide (LPS, 1 μg/mL, M1), and interleukin-4 (IL-4, 20 ng/mL, M2). The polarization markers analyzed included IL-6, NF-κB, NOS2, and STAT1 for the M1 phenotype and IL-10, VEGF, Arg-1, and STAT6 for the M2 phenotype, respectively.

### 4.2. Reverse-Transcription (RT)-qPCR Assay

Total RNA was purified from the cells using the Total RNA Isolation Kit (Blood/Cultured Cell/Fungus, GeneDireX, Taoyuan, Taiwan. Cat. No. NA017-0100) according to the manufacturer’s instructions. A GScript First-Strand Synthesis Kit (GeneDireX, Cat. No. MB305-0050) was used to synthesize cDNA from total RNA. The quantitative real-time polymerase chain reaction (qPCR) was performed using TaqMan™ Universal Master Mix II (no UNG, Applied Biosystems, Waltham, MA, USA. Cat. No. 4440040) and TaqMan probes (CD32b, Applied Biosystems, Cat. No. 4331182, ID Mm00438875_m1) following the manufacturer’s guidelines (QuantStudio™ 5 Real-Time PCR System, Applied Biosystems). GAPDH (TaqMan probe, Cat. No. 4331182, ID Mm99999915_g1) was used for standard curve preparation. The ΔCt value was estimated by using the reference gene as the internal calibration point, and then calibration was performed. ΔΔCt, corresponding to the reference point, and the relative gene expression were calculated using 2^−ΔΔCt^ values for all the samples.

### 4.3. Nitric Oxide (NO) Detection

RAW264.7 cells and primary macrophages from the spleens (2 × 10^5^ cells/mL) were cultured with the addition of phosphate-buffered saline (PBS, control), lipopolysaccharide (LPS, 1 μg/mL), and interleukin-4 (IL-4, 20 ng/mL) for 48 h. Then, the cell-free conditioned medium was collected, and NO production was measured by quantifying the nitrate/nitrite levels with Griess reagent and spectrophotometric detection at the wavelength of 540 nm.

### 4.4. Surface Marker Assay of CD32b

The treated cells were collected and reacted with anti-CD32b (1:100, Senta Cruze, Heidelberg, Germany) primary antibody, and then the immunofluorescence PE-conjugated anti-IgG-TR antibody (1:500, Santa Cruz) was hybridized to detect the expression level at the cell surface. The percentage of positive cells was detected using a NucleoCounter NC-3000 system (ChemoMetec A/S, Allerød, Denmark) and analyzed by using FCS Expression software (Version 4.0, DeNovo Software, Version 4.0, Los Angeles, CA, USA).

### 4.5. CD32b siRNA Transfection and RT-qPCR Assay of M1/M2 Markers

The Raw264.7 and primary macrophages isolated from mouse spleens were cultured in appropriate cell culture media for 24 h, and then we collected the cells for CD32b siRNA (Silencer^®^ Select siRNA, No. 4390771, ID s65920, Invitrogen, Thermo Fisher Scientific, Waltham, MA, USA) transfection by using Lipofectamine^®^ RNAiMAX Reagent (Invitrogen, Cat. No. 13778075). The control siRNA (Silencer™ Select Negative Control No. 1 siRNA, Invitrogen, Cat. No. 4390843) was also used for transfection to exclude interference and confirm the expression of CD32b. After transfection, the cells were cultured for further 24 h. Total RNA was extracted from the cells using a Total RNA Isolation Kit (Blood/Cultured Cell/Fungus, GeneDireX, Cat. No. NA017-0100) according to the manufacturer’s instructions. The isolated total RNA was reverse transcribed into cDNA by using a GScript First-Strand Synthesis Kit (GeneDireX, Cat. No. MB305-0050), and then we performed a qPCR assay. The analysis method was conducted as described previously (Section 4.2). For the analysis of the M1 and M2 markers, the primer sequences used for qPCR are listed in Table 1.

### 4.6. Human Subjects and Cytopathological Section Assay

Pleural effusion and ascites samples of patients with cancer were obtained from the Department of Pathology, National Taiwan University Hospital, Yunlin Branch, Taiwan. This study was approved by the Ethics Committee of National Taiwan University Hospital Yunlin Branch (No. 202207135RIN). This study was conducted in a retrospective manner, and the clinical pathology diagnosis tool was PATHOLOGIC STAGE CLASSIFICATION (pTNM, AJCC). The information in the samples included age, gender, tumor location, pathologic staging, distant metastasis, etc. The collected information of 32 patients with cancer who developed pleural effusion and ascitic fluid, including tumor stage T0-T1 (*n* = 6) and T2-T4 (*n* = 26), as well as non-metastasis (*n* = 6) and distant metastasis (*n* = 26), is listed in Table 2. The cytopathological specimens were fixed in 10% formalin solution, and through gradual alcohol dehydration and xylene transparency, the moisture in the specimens was completely replaced with paraffin, and then an appropriate amount of paraffin was added to cover the specimens. After cooling and solidification, we used a Leica HistoCore Biocut microtome to cut out 4 μm continuous tissue slices and attach them to glass slides. The specimens on the slides were stained with Hematoxylin and Eosin (H&E), and Immunohistochemistry (IHC) was performed. The primary antibody included anti-CD32b (Abcam, Cambridge, UK), anti-CD19 (Leica Biosystem, Nussloch, Germany), and anti-CD68 (Abcam). The slides were observed under the 200× microscopic field of view, and the images were captured with a Microscope Digital Camera DP27 (Olympus, Tokyo, Japan). Five fields of view were captured for each IHC staining label, and the DAB-labeled positive color percentage was analyzed using ImageJ 1.54g (Image Processing and Analysis in Java, NIH, Bethesda, MD, USA).

### 4.7. Statistical Analysis

All statistical analyses were carried out using SigmaStat software, Version 4.0 (Jandel Scientific, San Rafael, CA, USA). The results are expressed as the mean ± standard deviation (SD) from at least three independent experiments. All the experiments were compared among the different groups with Student’s *t*-test or analysis of variance, and the differences were considered significant at *p* < 0.05.

## 5. Conclusions and Future Directions

The present study is the first one to demonstrate the dynamic regulation of CD32b expression during macrophage polarization, implicating its involvement in M1 suppression and M2 polarization. Elevated M2 activation and CD32b levels may contribute to tumor progression, suggesting the potential therapeutic relevance of CD32b inhibition in attenuating M2 macrophage polarization and exerting anti-tumor immunity.

## Figures and Tables

**Figure 1 ijms-25-09737-f001:**
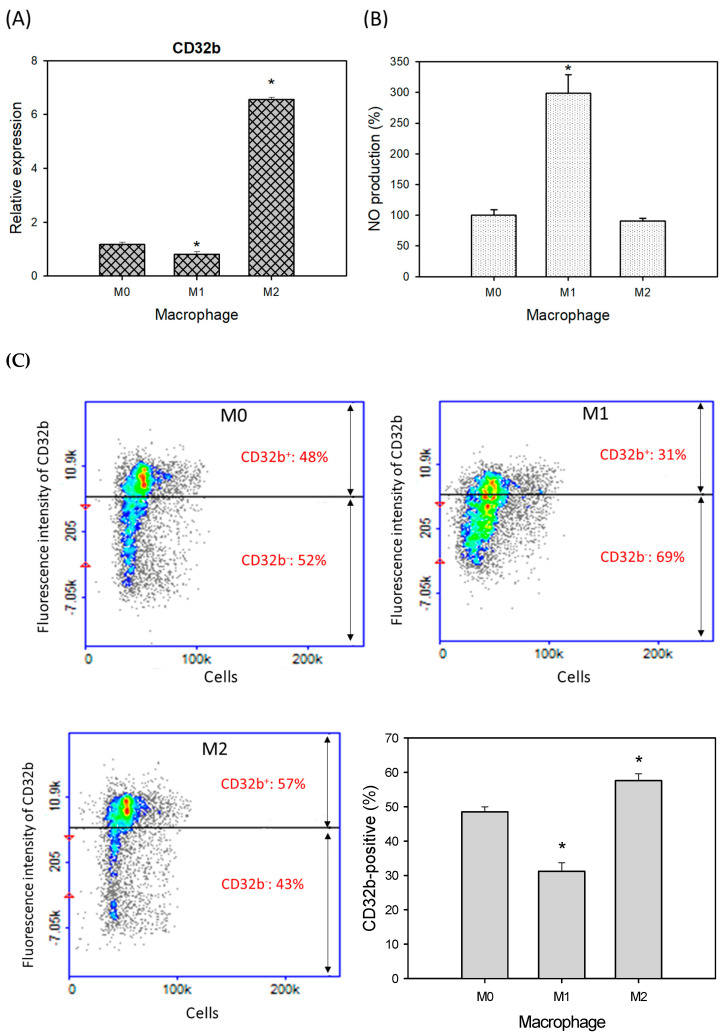
CD32b expression in macrophage Raw264.7 cells with treatments of PBS (M0), LPS (1 μg/mL, M1), and IL-4 (20 ng/mL, M2). (**A**) CD32b gene expression determined by qPCR assay, (**B**) nitric oxide (NO) production, and (**C**) cell surface antigen expression of CD32b. * Results are expressed as mean ± SD from three independent experiments, and differences are considered significant at *p* < 0.05.

**Figure 2 ijms-25-09737-f002:**
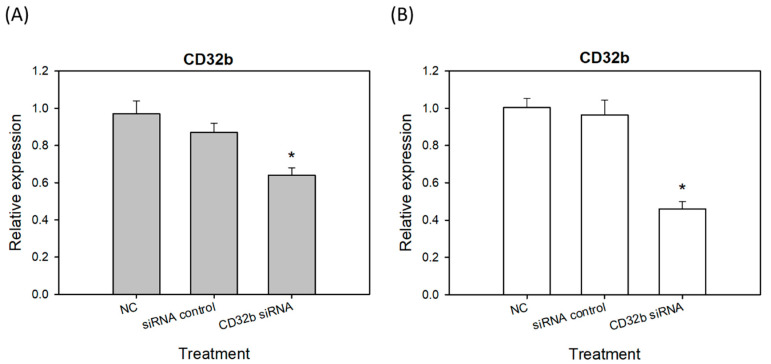
The CD32b expression in (**A**) the macrophage Raw264.7 cells transfected with CD32b siRNA and (**B**) the primary macrophages from the mouse spleens transfected with CD32b siRNA. * The results are expressed as the mean ± SD from three independent experiments, and the differences are considered significant at *p* < 0.05.

**Figure 3 ijms-25-09737-f003:**
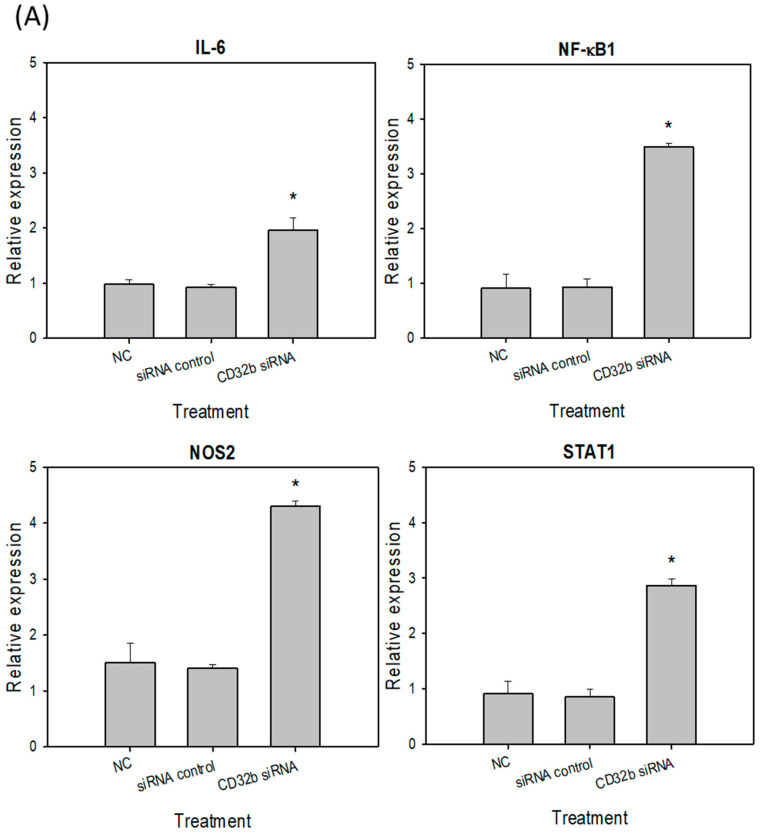
M1 marker (L-6, NF-κB, NOS2, and STAT1) expression in (**A**) macrophage Raw264.7 cells and (**B**) primary macrophages from mouse spleens transfected with CD32b siRNA and assayed by qPCR. * Results are expressed as mean ± SD from three independent experiments, and differences are considered significant at *p* < 0.05.

**Figure 4 ijms-25-09737-f004:**
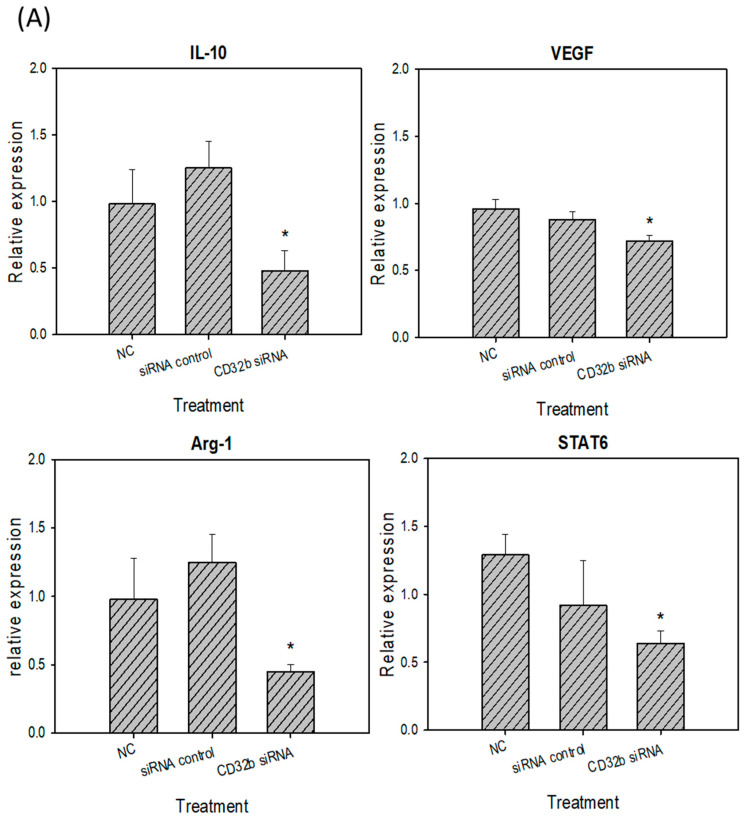
M2 marker (IL-10, VEGF, Arg-1, and STAT6) expression in (**A**) macrophage Raw264.7 cells and (**B**) primary macrophages from mouse spleens transfected with CD32b siRNA and assayed by qPCR. * Results are expressed as the mean ± SD from three independent experiments, and differences are considered significant at *p* < 0.05.

**Figure 5 ijms-25-09737-f005:**
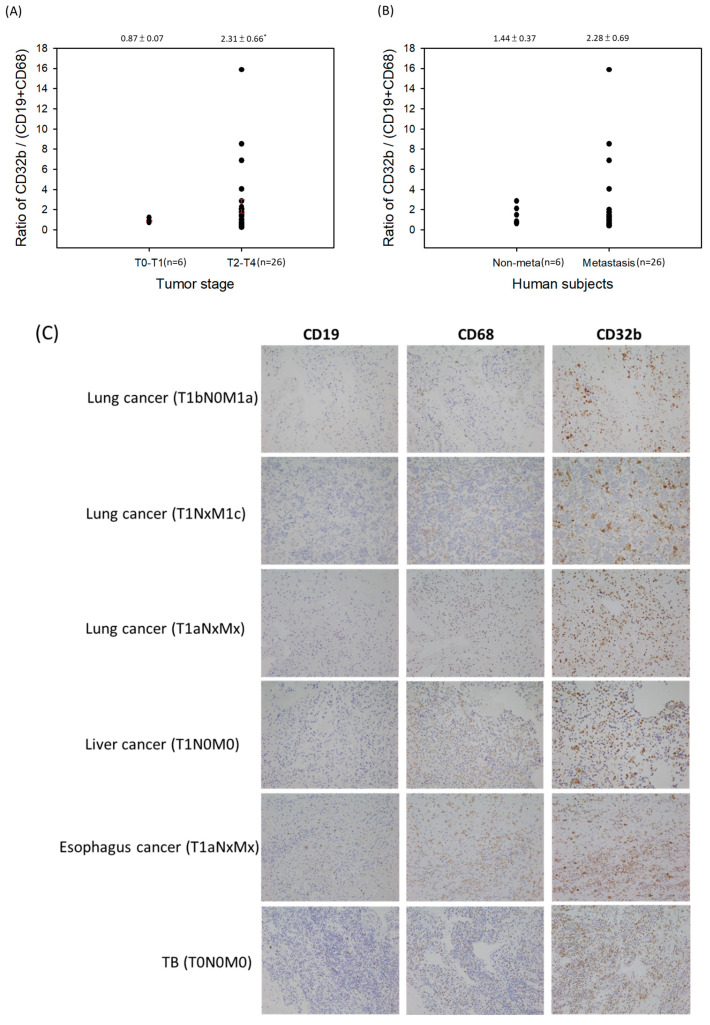
The ratio of CD32b/(CD19+CD68) in the cytopathological sections of the patients with cancer’s pleural effusion and ascitic fluid by IHC staining. (**A**) A comparison of the tumor stages T0–T1 and T2–T4. (**B**) A comparison of the non-metastasis and distant metastasis of tumors. (**C**) The IHC staining of sections from stages T0–T1. (**D**) The IHC staining of the sections selected from stages T2–T4. * The results are expressed as the mean ± SD from three independent experiments, and the differences are considered significant at *p* < 0.05. Each point on Figure 5A,B is the data of each subject, and the red line is the value of mean ± SD.

**Table 1 ijms-25-09737-t001:** Primers used for qPCR assay of M1 and M2 markers.

Genes	ID	Primer Sequences
STAT1	U06924.1	5′-TTCAGCAGCTGGACTCCAAG-3′ 3′-CGAGACATCATAGGCAGCGT-5′
NF-κB1	NP_032715.2	5′-AAGTGATCCAGGCAGCCTTC-3′ 3′-CTGTCACAGACGCTGTCACT-5′
NOS2	NM_010927.3	5′-CTATGGCCGCTTTGATGTGC-3′ 3′-TTGGGATGCTCCATGGTCAC-5′
IL-6	NM_031168.1	5′-GCCTTCTTGGGACTGATGCT-3′ 3′-AGCCTCCGACTTGTGAAGTG-5′
STAT6	NM_009284.2	5′-AGATGGGACCTTCCTCCTCC-3′ 3′-CTGAGCAAGATCCCGGATCC-5′
Arg-1	NM_007482.3	5′-TACATTGGCTTGCGAGACGT-3′ 3′-ATCACCTTGCCAATCCCCAG-5′
IL-10	NM_010548.2	5′-CAGAGAAGCATGGCCCAGAA-3′ 3′-GCTCCACTGCCTTGCTCTTA-5′
VEGFa	AAH61468.1	5′-AACGATGAAGCCCTGGAGTG-3′ 3′-CTGCTGTGCTGTAGGAAGCT-5′

**Table 2 ijms-25-09737-t002:** Information of patients with cancer with pleural effusion and ascites fluid collected in this study.

Tumor Stage (n)	T0–1 (6)	T2–T4 (26)
Age range (yr)	52–89	46–92
Pathological site (n)	Lung (3)	Lung (22)
	Liver (1)	Esophagus (1)
	Esophagus (1)	Ovarian (3)
	TB (1)	
Gender (F, M)	F (2)	F (13)
	M (4)	M (13)
**Non-Metastasis and metastasis of Tumors (n)**	**Non-metastasis (6)**	**Metastasis (26)**
Age range (yr)	56–83	46–92
Pathological site (n)	Lung (4)	Lung (21)
	Esophagus (1)	Esophagus (1)
	TB (1)	Ovarian (3)
		Liver (1)
Gender (F, M)	F (1)	F (14)
	M (5)	M (12)

## Data Availability

Data are contained within the article.

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
