# Peer review of "The Immunosuppressive Receptor CD32b Regulation of Macrophage Polarization and Its Implications in Tumor Progression"

_ijms, 2024, doi:10.3390/ijms25179737_

Round 1
Reviewer 1 Report
Comments and Suggestions for Authors
The authors induced CD32b in macrophages in cell culture in vitro and, using the human metastatic pleural effusion/ascites, proved that CD32b in tumor cells in the cito block is significantly associated with higher stage and worse prognosis.
The consistency of the performed methods is relevant (receptor induction in M 0, transfection, analysis of surrounding cells for induction cytokines, metastatic effusions from human cancer) and correct steps in proving the role of CD32b.
It is the consistency of evidence that fills a specific gap in research into the role and significance of CD32b.
The authors emphasized the fact that CD32b is involved in resistance to Her-2 blockade in breast cancer therapy, which is very good goal for further research on CD32b in Her-2 positive tumors.
The immunohistochemical expression of CD32b should be examined in solid tumors of different organs and correlated with prognostic parameters; chronic inflammation of that organ should be taken as a control.
The conclusions are in accordance with the presented evidence and arguments. References are appropriate.
Figure 2 needs to be better explained, as well as the measurement of CD32b concentration which correlates with a higher stage of disease. A better description of qPCR siRNA for readers unfamiliar with this method, along with an image. 2 which needs to be better explained.
Author Response
Comments 1: The authors induced CD32b in macrophages in cell culture in vitro and, using the human metastatic pleural effusion/ascites, proved that CD32b in tumor cells in the cito block is significantly associated with higher stage and worse prognosis.
The consistency of the performed methods is relevant (receptor induction in M 0, transfection, analysis of surrounding cells for induction cytokines, metastatic effusions from human cancer) and correct steps in proving the role of CD32b.
It is the consistency of evidence that fills a specific gap in research into the role and significance of CD32b.
The authors emphasized the fact that CD32b is involved in resistance to Her-2 blockade in breast cancer therapy, which is very good goal for further research on CD32b in Her-2 positive tumors.
The immunohistochemical expression of CD32b should be examined in solid tumors of different organs and correlated with prognostic parameters; chronic inflammation of that organ should be taken as a control.
The conclusions are in accordance with the presented evidence and arguments. References are appropriate.
Response 1: Thank you for the valuable suggestions. Our response and correction instructions are as follows:
We add the following explanation in the results section on page 16: The expression of CD19 (B cells) and CD68 (macrophages) in the specimen showed that the T2-T4 stage of the tumor was higher than that of T0-T1, indicating that immune cells increased in the later stages of the tumor. However, the high expression of CD32b in the T2-T4 stage (Figure 5D) indicates that immune cells have initiated the CD32b-mediated inhibitory signaling pathway, which may cause the anti-tumor immunity to be blocked.
At the same time, we add the following explanations in the discussion section on pages 20: Therefore, it is important to reconfirm in future studies, including the molecular mechanism of CD32b signaling pathway in macrophages and other immune cells, and clinical immunohistochemical expression of CD32b in solid tumors of different organs, to further analyze its correlation with prognostic parameters and metastasis, and to use chronic inflammation of the organ as a control.
Comments 2: Figure 2 needs to be better explained, as well as the measurement of CD32b concentration which correlates with a higher stage of disease. A better description of qPCR siRNA for readers unfamiliar with this method, along with an image. 2 which needs to be better explained.
Response 2: We add the following explanation in Figure 2 on page 13: Figure 1 has proven that CD32b regulates macrophages to move towards M2 polarization. Therefore, siRNA technology is further used to inhibit CD32b and further confirm the role of CD32b in macrophage polarization. In Figure 2, CD32b siRNA was transfected to macrophage Raw264.7 cell line and primary macrophages from the mouse spleen. Using RT-qPCR to detect the relative expression of CD32b, Raw264.7 cells transfected with CD32b siRNA can significantly increase the expression of CD32b mRNA (Figure 2A).
The method of reverse-transcription (RT)-qPCR assay was revised in section 2.2, and the CD32b siRNA transfection method was explained in section 2.5.
Reviewer 2 Report
Comments and Suggestions for Authors
Authors suggested the decrease in CD32b levels in LPS-treated M1 and an increase in IL-4 treated M2 macrophages in macrophage Raw264.7 cells along with the effect of CD32b depletion on the markers of M1 and M2. Despite interesting data, it has some concerns as follows:
1. Check English expression and grammar. Title, the relative expression of CD32b were decreased; studies have observed significantly reduced CD32b expression on macrophages in autoimmune disease patients
2. Show the effect of CD32b depletion on markers of M1 and M2, since we cannot always trust RT-PCR data
3. Can you confirm no effect on CD19 and CD68 in lung or liver cancer tissues?
4. Show the effect on CD206 in human cancer tissues
5. Discuss the relationship between sFGL2 and CD32B
6. Effect of CD32B depletion on sFGL2 and vice versa
7. Add size in IHC pictures and make bar graph with quantified values in at least 3 smaples
8. There are too much of Introduction part. Discuss the data carefully
9. How did you isolate macrophages in spleen of mice? Add method
Comments on the Quality of English LanguageCheck English expression and grammar. Title, the relative expression of CD32b were decreased; studies have observed significantly reduced CD32b expression on macrophages in autoimmune disease patients
Author Response
Authors suggested the decrease in CD32b levels in LPS-treated M1 and an increase in IL-4 treated M2 macrophages in macrophage Raw264.7 cells along with the effect of CD32b depletion on the markers of M1 and M2. Despite interesting data, it has some concerns as follows:
Comments 1: Check English expression and grammar. Title, the relative expression of CD32b were decreased; studies have observed significantly reduced CD32b expression on macrophages in autoimmune disease patients
Response 1: We have thoroughly revised the manuscript to improve the English expression and grammar. Thanks a lot.
Comments 2: Show the effect of CD32b depletion on markers of M1 and M2, since we cannot always trust RT-PCR data
Response 2: It’s true we cannot always trust RT-PCR data. Due to previous published studies (references 8-15) have proven that CD32b is a receptor on immune cells that provides immune-suppressive signals. CD32b deficiency is associated with autoimmune diseases, and CD32b overexpression is associated with cancer. This study is the first to demonstrate the role of CD32b in regulating macrophage polarization. Further research can be done in the future to reconfirm the results, including the immunoregulation of CD32b in solid tumors of different organs, which can be further analyzed and correlated with prognostic parameters and metastasis, as well as the chronic inflammation of the organ can be used as a control.
Comments 3: Can you confirm no effect on CD19 and CD68 in lung or liver cancer tissues?
Response 3: In Figure 5, the expression of CD19 (B cells) and CD68 (macrophages) in the specimen showed that the T2-T4 stage of the tumor was higher than that of T0-T1, indicating that immune cells increased in the later stages of the tumor. However, the high expression of CD32b in the T2-T4 stage (Figure 5D) indicates that immune cells have initiated the CD32b-mediated inhibitory signaling pathway, which may cause the anti-tumor immunity to be blocked.
This study initially observed that the IHC staining of cytopathological sections in patients with pleural effusion and ascitic fluid. We believe that future studies of lung or liver tumor tissue can further confirm the role of CD32b.
Comments 4. Show the effect on CD206 in human cancer tissues
Response 4: We added the effect of CD206 in Discussion (page 20): Modak et al. (2022) showed that alternatively activated CD206+ M2a macrophages efficiently cross-present soluble tumor-associated antigens (TAA), which is leading to tumor antigen-directed cytotoxic CD8+ T cell responses and type I IFN signaling, a key aspect of antitumor immunity [32].
Comments 5: Discuss the relationship between sFGL2 and CD32B
Response 5: We explained the relationship between sFGL2 and CD32b in Discussion (page 20): Higher expression of regulatory T cells (Tregs) -derived soluble fibrinogen-like protein 2 (sFGL2) and its receptor CD32b may induce macrophages toward a pro-repair phenotype via SHP2-ERK1/2-STAT3 signaling pathway, which is involved in the progression of endometriosis [31].
Comments 6: Effect of CD32B depletion on sFGL2 and vice versa
Response 6: According to the research by Hou et al. (Commun. Biol., 2021), Tregs secrete sFGL2, which then acts on the receptor CD32b to regulate macrophage M2 polarization.
The macrophages treated with rFGL2 expressed more pro-repair markers CD163 and CD206, which was accompanied by an increase in the levels of the anti-inflammatory cytokines TGF-β and IL-10. The anti-CD32B neutralizing antibody pretreatment before adding rFGL2 blocked such M2 effects, suggesting that sFGL2-induced pro-repair macrophages polarization is CD32b-dependent.
The macrophages co-cultured with Tregs also showed increased levels of CD163, CD206 expression and TGF-β, IL-10 production. The anti-FGL2 antibody partly blocked these effects, showing that other factors may also affect macrophage polarization.
Comments 7: Add size in IHC pictures and make bar graph with quantified values in at least 3 smaples
Response 7: We added the size in IHC pictures in Figure 5D. Expression quantification was performed on all samples from T0-T1 (n = 6), T2-T4 (n = 26), non-metastatic (n = 6) and metastatic (n = 26) patients, and the results are shown in Figures 5A and 5B.
Comments 8: There are too much of Introduction part. Discuss the data carefully
Response 8: We revised the introduction part and discussed the data carefully.
Comments 9. How did you isolate macrophages in spleen of mice? Add method
Response 9: We add the following explanation in the method section on page 7-8: The mice were sacrificed and the spleen organs were removed. RPMI 1640 medium 10 mL was added to the spleen organ, homogenized the tissue into single cells, and dispersed in suspension by using a 1-mm metal sieve. The isolated spleen cells (5 × 105 cells/mL) were cultured at 37°C in RPMI 1640 medium containing 10% FBS for 2 h, and the adherent cells are macrophage-rich fraction. Use a rubber scraper to collect the cells and suspend them in the culture medium.
Comments on the Quality of English Language
Comments 10: Check English expression and grammar. Title, the relative expression of CD32b were decreased; studies have observed significantly reduced CD32b expression on macrophages in autoimmune disease patients.
Response 10: The English expression and grammar has been revised.
Round 2
Reviewer 2 Report
Comments and Suggestions for Authors
Much improved